# MolCoMA: Complementary Masking Strategy for Promoting Atom-Level Multi-Modal Molecular Representation

## Abstract

Molecular representation learning, which captures the fundamental characteristics of chemical compounds, is crucial for AI-driven drug discovery. Methodologies exist that integrate various modalities (e.g., 2D topology and 3D geometry) and develop robust representations. However, current multi-modal fusion strategies either align embedding space through independent models separately, thereby overlooking complementary information, or bridge modalities at a coarse-grained level, failing to capture inherent correlation. To facilitate fine-grained interactions of intrinsic features across modalities, this study presents MolCoMA, an innovative pretraining framework for **Mol**ecular representation, employing a unified encoder that leverages **Co**mplementary **Ma**sking mechanism. Specifically, we first employ two distinct encoders to capture the unique characteristics and structures inherent in different modalities. We then utilize a unified encoder accompanied by a customized complementary masking strategy to seamlessly integrate information, mitigating overlap and similarity between 2D and 3D representations. Finally, we incorporate a cross-modal reconstruction module to enhance fine-grained interactions at the atomic level. Extensive experiments demonstrate that our model outperforms existing molecular pretraining methods across both 2D and 3D benchmarks. This finding underscores the effectiveness of our approach to fusing information between modalities.

## 1 Introduction

Robust molecular representations serve as a cornerstone for AI-based models to ensure their successful application in drug discovery (Simeon & De Fabritiis, 2024; Song et al., 2024; Zhou et al., 2024). Molecular representations not only assist the models in precisely capturing the intricate intrinsic properties of molecules and gaining deeper insights into their mechanisms, but also provide a solid foundation for tackling a wide range of downstream tasks, including drug screening (Gao et al., 2024) , drug optimization (Zhu et al., 2024) and drug design (Lin et al., 2024). With the surge of unlabeled data, the utilization of pre-training models that leverage self-supervised learning to extract inherent molecular characteristics further enhances models' generalization ability and accuracy, presenting unprecedented opportunities and challenges for AI-driven drug discovery.

Molecules are represented in diverse modalities, encompassing 1D, 2D and 3D formats. Among these, 1D representations, devoid of structural information, often yield insufficient representations to support downstream tasks effectively. In contrast, 2D representations encapsulate the topological structure of a molecule, with atoms as nodes and bonds as edges. This captures intricate structural features and inter-atomic interactions that are crucial for deeper understanding and downstream applications (Hou et al., 2022; Yan et al., 2024; Rong et al., 2020). 3D representations encapsulate the 3D spatial coordinates of atoms, reflecting the crucial structural arrangement information that is paramount for comprehending molecule functionalities (Schütt et al., 2017; Ni et al., 2024b;a; Zaidi et al., 2023b). In contrast to previous studies, 2D methods, which concentrate solely on topological connections between atoms, overlook spatial information that is a domain where 3D modalities demonstrate their superiority. Different modalities showcase varying advantages through offering diverse perspectives and information, and they frequently exhibit complementarity. Consequently, relying solely on a single modality for learning molecular representations fails to comprehensively

capture molecular properties, making it difficult to develop robust representations that are adaptable to a wide range of downstream tasks.

Currently, existing approaches aim to adopt multi-modal learning to integrate information spanning a variety of sources, thereby achieving a more comprehensive understanding (Li et al., 2022; Liu et al., 2022; 2023a; Stärk et al., 2022; Luo et al., 2022; Zhu et al., 2022). The key to multi-modal learning is to interact and align various information, enabling models to handle complex relationships between different modalities. As shown in Figure 1, there are two prevailing strategies to tackle this kind of problem: interaction-based and generation-based. Interaction-based strategy emphasizes the integration of feature representations from different modalities, thereby learning interaction relationships between multi-modal data. For instance, contrastive learning is employed to align various modal information of the same molecule within the embedding space (Liu et al., 2022; 2023a; Stärk et al., 2022). In contrast, the generation-based approach stresses that one modality serves as a constraint to construct a bridge to the space of another modality, thereby enabling mutual data generation between modalities and capturing rich cross-modal information (Liu et al., 2023a; Zhu et al., 2022). This method often utilizes generative models to achieve mutual reconstruction as the goal, such as the transformation from 2D to 3D modalities (or vice versa). The former emphasizes the consistency between modalities but fails to effectively capture complex correlations or dependencies between two modalities; while the latter places more emphasis on the quality of the generated modality rather than directly enhancing the information interaction between modalities, unable to ensure that the interaction between modalities is fully optimized. Therefore, it is crucial to explore how to achieve interaction relationship mining at a fine-grained level between different modalities.

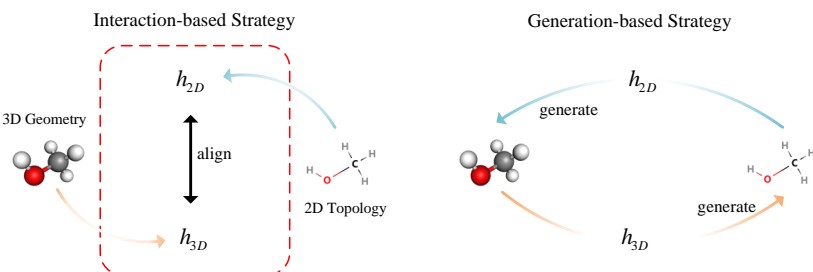

Figure 1: Illustration of two prevailing multi-modal learning strategies: Interaction-based (left) and Generation-based (right). The former aims to align embedding space of different modalities, which usually adopts contrastive learning method. The latter relies on bridging two modalities via mutual construction.

To address the aforementioned issue, we propose MolCoMA, a straightforward yet effective multi-modal pretraining framework that employs a complementary masking strategy and cross-modal reconstruction on 2D topology and 3D geometry at a fine-grained level. Unlike conventional uniform or random masking strategies, our approach features a well-designed masking technique that obscures atoms across different modalities in a complementary manner. The corrupted multi-modal data is subsequently fed into two specialized modality-specific encoders, which independently and efficiently extract the intrinsic representations of each modality. This process aims to enhance the model's capacity to understand and represent single-modality features. Subsequently, a unified encoder is utilized to derive unified representations from both modalities by replacing the embeddings of masked atoms with their corresponding unmasked embeddings from the other modality. Ultimately, these unified representations are partitioned into two distinct embeddings associated with different modalities by recovering masks at their respective indices; these are then separately inputted into specific decoders for reconstructing both original and cross-modal information. Comprehensive experiments demonstrate that our model exhibits significant advantages over existing molecular representation methods, irrespective of whether it utilizes 2D, 3D, or multi-modal information across an extensive range of benchmarks.

Our contributions are summarized as follows:

• We present MolCoMA, an innovative molecular pretraining framework that leverages the complementary information from topology and geometry representations while facilitating cross-modal interaction, ultimately leading to enhanced performance.

• We propose a straightforward yet effective complementary masking strategy, along with a unified backbone and cross-modal reconstruction approach for addressing two distinct modalities. This framework facilitates fine-grained interactions between the modalities.

• Through comprehensive experiments on biological tasks in MoleculeNet and quantum tasks in QM9, our model outperforms existing molecular pretraining methods, demonstrating its superior performance.

## 2 RELATED WORK

**Single modal molecular pretraining.** Pretraining on 2D molecule topology shares similar concepts with general graph pretraining (Wang et al., 2024). A traditional approach (Hu et al., 2020; Liu et al., 2019) involves masking specific substructures in molecule graphs, such as atoms, bonds, or motifs, and then reconstructing them using an auto-encoding method. GraphMAE (Hou et al., 2022) performs feature reconstruction of masked atoms rather than explicitly reconstructing graph structures and achieves superior performance. Another common method for molecular pretraining is contrastive learning (Oord et al., 2018), where the objective is to simultaneously align views from positive pairs and contrast views from negative pairs(Luong & Singh, 2024). For example, Deep Graph InfoMax (Veličković et al., 2019) and InfoGraph (Sun et al., 2019) consider local and global graph representations as two different views, while MolCLR (Wang et al., 2022) and GraphCL (You et al., 2020a) generate distinct views using discrete graph augmentation techniques.

There are also several existing works exploring 3D geometric pretraining on molecules. SE(3)-DDM (Liu et al., 2023b) proposes maximizing mutual information between noisy conformations by employing an SE(3)-invariant denoising score matching approach. 3D-EMGP (Jiao et al., 2023) is a parallel work that is E(3)-equivariant and inherently meets the reflection-equivariant constraint in the distribution of molecule conformations.

**Multiple modal molecular pretraining.** Multi-modal molecular pretraining (Luo et al., 2022; Zhu et al., 2022; Liu et al., 2023a; 2022; Stärk et al., 2022) utilizes both 2D and 3D information to enhance molecule representation learning. Most existing methods employ two separate models to encode 2D and 3D information (Liu et al., 2022; 2023a; Stärk et al., 2022). Their pretraining techniques primarily use contrastive learning (He et al., 2020), where 2D graphs and their corresponding 3D conformations are treated as positive views, while information from different molecules serves as negative views. Another pretraining method uses generative models to predict one modality based on the input of another modality (Liu et al., 2023a; 2022). However, these models only achieve alignment at the molecular level and lack fine-grained interactions between modalities, missing a deep comprehension of molecular structure.

**Masked graph modeling.** Masked graph modeling (MGM) (Hou et al., 2022; Xia et al., 2023; You et al., 2020b) consists of three key components: graph tokenizer, graph masking, and graph autoencoder. Graph tokenizer (Zhang et al., 2021; Rong et al., 2020; Sun et al., 2021) breaks a molecular graph into smaller fragments (i.e., subgraphs) and converts them into tokens. Graph masking (Hu et al., 2020; You et al., 2020b) creates a corrupted graph with masks. Graph autoencoder (Hu et al., 2020; Hou et al., 2022) applies an encoder to the masked graph to generate representations, and then employs a decoder on the representations to recover the tokens of the original graph. Early works utilized the atomic numbers of nodes and bond types of edges as graph tokens largely due to their simplicity (You et al., 2020b; Hu et al., 2020). Recent studies (Jin et al., 2018; Zhang et al., 2021) have shifted focus to subgraph-level tokenization, such as motifs (Salmina et al., 2015) and pretrained GNNs (Xia et al., 2023).

## 3 METHOD

In this section, we introduce MolCoMA, a multi-modal molecular pretraining framework designed to facilitate interactions between 2D and 3D representations. We begin by presenting the formulation of the problem. Subsequently, we illustrate a straightforward yet effective complementary masking

strategy. Following this, we detail our meticulously designed encoders and decoders. Furthermore, we incorporate a cross-modal reconstruction module to enhance the interaction between different modalities even further.

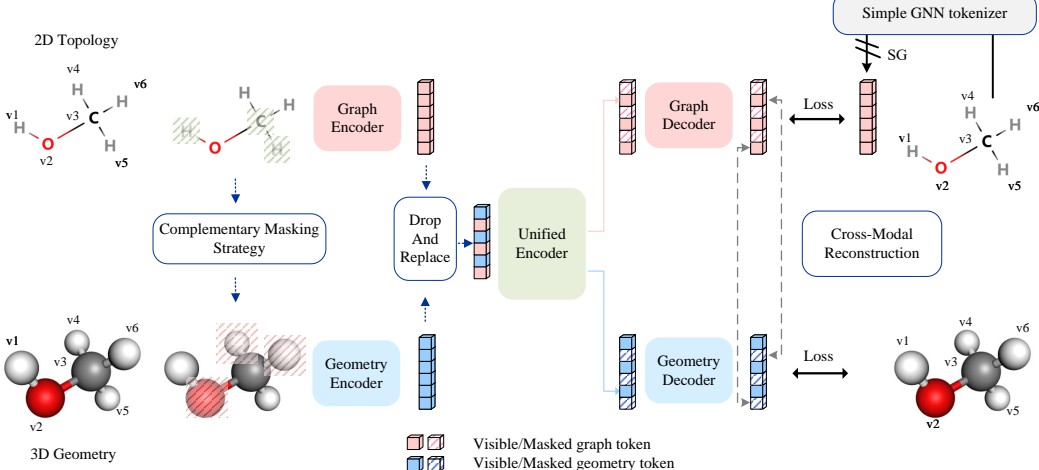

Figure 2: Pre-training pipeline for MolCoMA. The point cloud and graph are masked in a complementary manner, after which both modalities are processed through their respective encoders. Following the embedding stage, tokens undergo a unified encoder to facilitate cross-modal interaction. Finally, we utilize three decoders to achieve both intra-modal and inter-modal reconstruction. The abbreviation SG denotes stop gradient.

### 3.1 PROBLEM FORMULATION

A molecule $\mathcal{M}$ consists of a collection of atoms held together by attractive forces. The atoms with features can be denoted as $X \in \mathbb{R}^{n \times d}$, where $n$ is the number of atoms, and d is the feature dimension. The structure of $\mathcal{M}$ can be represented as different forms, such as 2D graph structures and 3D geometric structure. For the 2D graph structure, atoms serve as nodes and covalent bonds as edges. We define $\mathcal{M}_{2D} = (\boldsymbol{X}, \boldsymbol{E})$, where $\mathbf{e}_{(i,j)} \in \boldsymbol{E}$ denotes the edge feature (the type of the bond) between atom i and j if the edge exists. For the 3D geometric structure, atoms are treated as the point clouds in 3D Euclidean space. We define $\mathcal{M}_{3D} = (\boldsymbol{X}, \boldsymbol{R})$, where $\boldsymbol{R} = \{\mathbf{r}_1, ..., \mathbf{r}_n\}, \mathbf{r}_i \in \mathbb{R}^3$. We aim to design a powerful model which take $\mathcal{M}_{2D}$ and $\mathcal{M}_{3D}$ as input, obtain robust representation via incorporating complementary information between modalities.

### 3.2 COMPLEMENTARY MASKING STRATEGY

Existing multi-modal pretraining methods focus extensively on model architecture and pretraining objective, while there is limited exploration of multi-modal masking strategy (Liu et al., 2022; 2023a). As shown in Figure3, most works apply either a uniform masking or random masking strategy to the two modalities. Specifically, they concatenate tokens from different modalities into a sequence, then predict the masked tokens (Wang et al., 2023; Chen et al., 2020). This may result in feature redundancy due to the overlap or similarity between the two types of information, particularly in lower-level representations of molecular structures.

In molecular pretraining, 2D and 3D information each possess unique representational strengths. The 2D topology captures atomic connections and bond information, while the 3D structure represents the spatial conformation and geometric properties of molecules. By employing complementary masks, various parts of the information from the 2D and 3D modalities can be obscured, compelling the model to learn latent representations from one modality using the other. When information from one modality is hidden, the model must rely on the other modality to fill in the gaps. This allows the model to utilize the distinct characteristics of both modalities during pretraining, facilitating the integration of complementary information. As a result, the model gains a deeper understanding of the

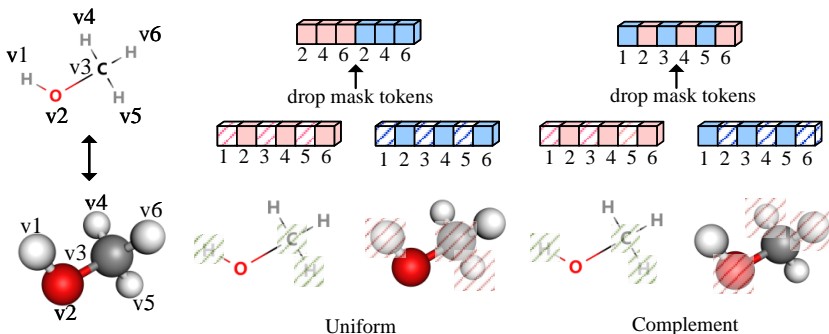

Figure 3: Comparison of the uniform masking approach (left), which is commonly employed by most methods, and our complementary masking strategy (right).

relationship between 2D and 3D data, leading to more robust molecular representations that enhance performance on downstream tasks.

In this work, we randomly mask atom features of the 2D topology (Liu et al., 2024), then transform the mask pattern onto the 3D geometry, where the coordinates of atoms are complementarily masked (Chen et al., 2023). Since the positions are continuous values, not discrete values, we cannot use a special value to represent the mask (Zhou et al., 2023). Thus, we add noise to the ground-truth positions rather than masking to corrupt the 3D positions, and the model is trained to predict noise. The mask ratio is set to 0.5.

## 3.3 ENCODING PHASE

**Modality-specific Encoder.** At this stage, our goal is to preserve the integrity of each modality, maintaining the unique characteristics and structure of the data from different modalities. This guarantees that the distinctive features of each modality remain intact throughout the processing pipeline, allowing for more accurate and effective multi-modal interactions in subsequent stages. We utilize two separate encoders to transform data from two distinct modalities into embeddings of the same dimensionality, allowing further processing.

For the 2D topological molecular graph, it can be denoted as $\mathcal{M}_{2D} = (\boldsymbol{X}, \boldsymbol{E})$, where $\boldsymbol{X}$ is the atom attribute matrix and $\boldsymbol{E}$ is the bond attribute matrix. We feed them into the 2D Graph Encoder:

$$\boldsymbol{H}_{2D} = \text{GNN-2D}(T_{2D}(\mathcal{M}_{2D})) = \text{GNN-2D}(T_{2D}(\boldsymbol{X}, \boldsymbol{E})), \tag{1}$$

where $T_{2D}$ is the data transformation on the 2D topology, and GNN-2D is the 2D Graph Encoder representation function. $\boldsymbol{H}_{2D} = [h_{2D}^0, h_{2D}^1, \ldots]$, where $h_{2D}^i$ is the i-th node representation. We take the GIN (Xu et al., 2019) model for modeling 2D topology, which performs exceptionally well in many tasks.

For the 3D geometric molecular graph, the conformation can be denoted as $\mathcal{M}_{3D} = (\boldsymbol{X}, \boldsymbol{R})$. The geometric representation is:

$$\boldsymbol{H}_{3D} = \text{GNN-3D}(T_{3D}(\mathcal{M}_{3D})) = \text{GNN-3D}(T_{3D}(\boldsymbol{X}, \boldsymbol{R})), \tag{2}$$

where $T_{3D}$ is the data transformation on the 3D geometry, and GNN-3D is the 3D Geometry Encoder representation function. $\boldsymbol{H}_{3D} = [h_{3D}^0, h_{3D}^1, \ldots]$, where $h_{3D}^i$ is the i-th node representation. We employ TorchMD-NET (Thölke & Fabritiis, 2022) as the 3D encoder, which is a novel Equivariant Transformer (ET) architecture.

**Unified Encoder.** We develop a unified Transformer-based backbone to perform feature interaction (Vaswani, 2017). By leveraging Transformer's attention mechanism, the model can effectively capture and learn the relationships between different input token embeddings across two modalities. The self-attention mechanism is formulated as:

$$\text{SelfAttention}(\boldsymbol{Q}, \boldsymbol{K}, \boldsymbol{V}) = \text{softmax}\left(\frac{\boldsymbol{Q}\boldsymbol{K}^T}{\sqrt{d}}\right)\boldsymbol{V}, \tag{3}$$

Note that, we remove the masked tokens and replace them with the corresponding unmasked token from the other modality, instead of simply contact tokens from different modalities. Due to complementary masking, various parts of the information from 2D and 3D modalities are obscured, which effectively avoids the overlap or similarity between the two types of representation, enabling a more comprehensive understanding and utilization of the combined data.

## 3.4 DECODING PHASE

**2D Decoder.** Recent studies (He et al., 2022) have revealed the relationship between representation learning and reconstruction tasks. The semantic level of the reconstruction target has effect on Decoder's design. In NLP, the targets are one-hot missing words with rich semantics, MLP is sufficient for decoding (Devlin et al., 2019). However, in CV, a more advanced decoder (e.g. Transformer model (Vaswani, 2017)) is required to recover pixel patches with low-level semantics (He et al., 2022). In graphs, since the reconstruction targets (e.g. node features) are usually less informative, a more expressive decoder is necessary. Following (He et al., 2022; Feichtenhofer et al., 2022) , we devise a smaller version of GraphTrans (Wu et al., 2021)(GTS) architecture as decoders that stack transformer layers on top of the GINE layers to improve the ability of modeling global interactions.

Apart from the decoder architecture, in MRL, the type of reconstruction target (e.g. Node, Motif, Frozen GNN representations) strongly impacts the encoder's representation learning as well (Zhang et al., 2022; Hou et al., 2022). Inspired by previous works (Liu et al., 2024), we employ a simplified GNN (Xu et al., 2019) to obtain the features of reconstruction targets (Simple GNN Tokenizer), in which the nonlinear update functions are removed. For a molecule, the features of target nodes to be reconstructed are shown as follows:

$$
\begin{aligned}
\mathbf{H}_i^{(0)} &= \text{Embedding}(\boldsymbol{x}_i) \in \mathbb{R}^d, \quad \forall i \in \mathcal{V}, \\
\hat{\mathbf{H}}(l) &= \omega(\mathbf{A}) \cdot \mathbf{H}^{(l-1)} \in \mathbb{R}^{|\mathcal{V}| \times d}, \quad 1 \le l \le k, \\
\mathbf{H}(l) &= \text{BatchNorm}(\hat{\mathbf{H}}^{(l)}), \\
\text{Target}(g) &= \{\mathbf{y}_i = \text{SG}([\mathbf{H}_i^{(1)}, ..., \mathbf{H}_i^{(k)}]) | i \in \mathcal{V}_m\},
\end{aligned}
\tag{4}
$$

where $\mathcal{V}$ denotes the set of nodes and $\mathcal{V}_m$ is the set of masked nodes; Embedding($\cdot$) is a linear layer that uses the weights of the encoder's node embedding function; $\mathbf{H}_i^{(l)}$ is the i-th row of $\mathbf{H}(l)$; BatchNorm($\cdot$) is a standard Batch Normalization layer without the trainable scaling and shifting parameters (Ioffe & Szegedy, 2015); $\omega(\mathbf{A})$ is the original GNN's aggregation function.

Our loss function, denoted as $\mathcal{L}_{2D}$, computes the mean squared error (MSE) between the reconstructed and target features calculated by Equation 4 . We compute the loss only on masked tokens.

**3D Decoder.** The goal of the 3D decoder is to predict the added noise based on the latent embeddings (Godwin et al., 2022; Zaidi et al., 2023a). Let $\boldsymbol{R} = \{\boldsymbol{r}_1, \boldsymbol{r}_2, \ldots\}, \boldsymbol{r}_i \in \mathbb{R}^3$ denotes the atom positions of a molecule, and the noisy version of the atom positions is $\boldsymbol{r}_i + \sigma\boldsymbol{\epsilon}_i, i \in \boldsymbol{R}_m$, where $\sigma$ is the scaling factor of noise, $\boldsymbol{R}_m$ denotes the set of corrupted positions. The prediction of the decoder is $\{\hat{\boldsymbol{\epsilon}}_1, ..., \hat{\boldsymbol{\epsilon}}_m\}$. In this work, we use the Transformer architecture as a decoder. The attention matrix is defined as:

$$
\boldsymbol{A}(\boldsymbol{X}) = \text{softmax}\left(\frac{\boldsymbol{X}\boldsymbol{W}_Q(\boldsymbol{X}\boldsymbol{W}_K)^\top}{\sqrt{d}}\right)
\tag{5}
$$

Following (Shi et al., 2022), we employ an SE(3) equivariant attention layer as the prediction head:

$$
\hat{\boldsymbol{\epsilon}}_i^k = \left(\sum_{v_j \in V} a_{ij} \Delta_{ij}^k \boldsymbol{X}_j^{(L)} \boldsymbol{W}_N^1\right) \boldsymbol{W}_N^2, \quad k = 0, 1, 2
\tag{6}
$$

where $\boldsymbol{X}_j^{(L)}$ is the output of the last Transformer block, $a_{ij}$ is the attention score between atom i and atom j calculated by Equation 5, $\Delta_{ij}^k$ is the k-th element of the directional vector $\frac{\mathbf{r}_i - \mathbf{r}_j}{\|\mathbf{r}_i - \mathbf{r}_j\|}$ between atom i and atom j, and $\boldsymbol{W}_N^1 \in \mathbb{R}^{d \times d}$, $\boldsymbol{W}_N^2 \in \mathbb{R}^{d \times 1}$ are learnable weight matrices. The loss of a batch of molecules $\mathcal{S} = \{V^1, ..., V^{|\mathcal{S}|}\}$ is the cosine similarity between the predicted noises and the ground-truth noises:

$$
\mathcal{L}_{3D} = \frac{1}{|\mathcal{S}|} \sum_{V^i \in \mathcal{S}} \sum_{j \in V_m^i} \left(1 - \cos\left(\boldsymbol{\epsilon}_j^i, \hat{\boldsymbol{\epsilon}}_j^i\right)\right)
\tag{7}
$$

**Cross-modal Reconstruction.** Apart from the topology and geometry loss terms, we additionally utilize a cross-modal reconstruction loss to train the model to further strengthen the fine-grained interaction between the two modalities. Compared to 3D representations, molecular 2D representations directly reflect the molecule's bonding relationships, providing important insights for predicting chemical properties. Moreover, 3D structures can be unstable due to conformational changes, thermal motion, or measurement errors, while 2D structures are relatively fixed, offering more stable and consistent representations. Thus, in this module, the masked geometric tokens go through a cross-modal prediction head of one linear projection layer to recover the corresponding visible topological graph tokens.

We avoid using visible geometric tokens for this module, because they correspond to masked graph features (due to the complement masking strategy), which tend to have weaker representations and may harm representation learning. Formally, the cross modal reconstruction loss is defined as:

$$\mathcal{L}_{cross} = MSE(D_{cross}(h_{2D}), h_{3D}), \tag{8}$$

where MSE denotes the Mean Squared Error loss function, $D_{cross}$ is the cross-modal reconstruction function, $h_{2D}$ is the graph representation, and $h_{3D}$ is the geometric representation.

The over loss of our model is the sum of the previous loss terms:

$$\mathcal{L}_{total} = \alpha_1 \cdot \mathcal{L}_{2D} + \alpha_2 \cdot \mathcal{L}_{3D} + \alpha_3 \cdot \mathcal{L}_{cross}. \tag{9}$$

where $\alpha_1, \alpha_2, \alpha_3$ are weighting coefficients. With such design, our model can learn 3D and 2D features independently while preserving strong fine-grained interactions between the two modalities.

## 4 EXPERIMENTS

### 4.1 EXPERIMENTAL SETTINGS

**Datasets.** We pretrain our model on the PCQM4Mv2 training set from OGB Large-Scale Challenge (Hu et al., 2021). It's a sub-dataset of PubChemQC (Nakata & Shimazaki, 2017), which contains 3.4 million molecules with both 2D topological graphs and 3D geometric conformations. With the pre-trained model, we conduct experiments on 20 molecular tasks with different data formats and evaluate its versatility and effectiveness. We study two representative tasks: MoleculeNet (Wu et al., 2018) (2D, 8 tasks), QM9 quantum properties (Ramakrishnan et al., 2014) (3D, 12 tasks).

**Baselines.** A range of 2D topological pretraining methods have been proposed. We pick the most representative ones as 2D baselines, including AttrMask (Hu et al., 2020; Liu et al., 2019), ContextPred (Hu et al., 2020), InfoGraph (Sun et al., 2019), MolCLR (Wang et al., 2022), GraphCL (You et al., 2020a), Mole-BERT (Xia et al., 2023) and GraphMAE (Hou et al., 2022). Moreover, we adopt several multi-modal pretraining methods, which aim to fusion 2D topology and 3D conformations, as multi-modal baselines, such as 3D infoMax (Stärk et al., 2022), GraphMVP (Liu et al., 2022), MoleculeSDE (Liu et al., 2023a), as well as recently published model MoleBlend (Yu et al., 2024). Moreover, we choose some denoising methods, SE(3)-DDM (Liu et al., 2023b) and 3D-EMGP (Jiao et al., 2023), as baselines of QM9 dataset.

**Pretraining and Fine-tuning.** We take the GIN (Xu et al., 2019) model as 2D encoder and TorchMD-NET (Thölke & Fabritiis, 2022) as 3D encoder. The model is pre-trained on PCQM4Mv2 for 100 epochs. We adopt the AdamW optimizer with a warm-up for 10 epochs. The peak learning rate is 1e-4 and the batch size is 512. The mask ratio is 0.5. In fine-tuning stage, for 2D tasks, we utilize the 2D specific encoder as well as the unified encoder as a 2D feature extractor. For 3D tasks, we only adopt the 3D encoder as a 3D feature extractor considering time and efficiency. Different heads are employed for different tasks: a simple 1-layer MLP head for MoleculeNet and an equivariant head for QM9. Detailed configurations are in Appendix A.1.

### 4.2 MOLECULAR PROPERTY PREDICTION TASK (2D)

We assess our model using MoleculeNet, a widely used benchmark for 2D molecular property prediction, encompassing a variety of molecular properties, from quantum mechanics and physical chemistry to biophysics and physiology. The evaluation follows the scaffold split method (Wu et al.,

2018) and we report the mean and standard deviation of results based on three different random seeds.

Table 1 represents the performance of our method across 8 biological classification tasks in MoleculeNet. Impressively, our approach attains state-of-the-art performance in 6 out of the 8 tasks, with large margins in certain cases (e.g., 85.2 v.s. 83.7 on Bace). Compared with other multi-modal methods (3D infoMax (Stärk et al., 2022), GraphMVP (Liu et al., 2022), MoleculeSDE (Liu et al., 2023a)), which utilize contrastive learning as one of the objectives to align molecules at the molecule level, the improvement observed in our method can be attributed to the atomic interaction between two modalities (2D topology and 3D geometry). Moreover, our method also surpasses all 2D baselines (as shown in the upper section of the table), highlighting that the integration of 3D information enhances molecular properties prediction.

Table 1: Results of 2D molecular property prediction tasks on MoleculeNet. The evaluation is Mean (and Standard Deviation) of the ROC-AUC score (higher is better) under scaffold splitting.

| Pretraining | BBBP↑ | Tox21↑ | ToxCast↑ | SIDER↑ | ClinTox↑ | MUV↑ | HIV↑ | Bace↑ | Avg↑ |
|---|---|---|---|---|---|---|---|---|---|
| AttrMask (2020) | 65.0±2.3 | 74.8±0.2 | 62.9±0.1 | 61.2±0.1 | **87.7±1.1** | 73.4±2.0 | 76.8±0.5 | 79.7±0.3 | 72.7 |
| ContextPred (2020) | 65.7±0.6 | 74.2±0.0 | 62.5±0.3 | 62.2±0.5 | 77.2±0.8 | 75.3±1.5 | 77.1±0.8 | 76.0±2.0 | 71.3 |
| GraphCL (2020) | 69.7±0.6 | 73.9±0.6 | 62.4±0.5 | 60.5±0.8 | 76.0±2.6 | 69.8±2.6 | 78.5±1.2 | 75.4±1.4 | 70.8 |
| InfoGraph (2020) | 67.5±0.1 | 73.2±0.4 | 63.7±0.5 | 59.9±0.3 | 76.5±1.0 | 74.1±0.7 | 75.1±0.9 | 77.8±0.8 | 71.0 |
| GROVER (2020) | 70.0±0.1 | 74.3±0.1 | 65.4±0.4 | 64.8±0.6 | 81.2±3.0 | 67.3±1.8 | 62.5±0.9 | 82.6±0.7 | 71.0 |
| MolCLR (2022) | 66.6±1.8 | 73.0±0.1 | 62.9±0.3 | 57.5±1.7 | 86.1±0.9 | 72.5±2.3 | 76.2±1.5 | 71.5±3.1 | 70.8 |
| GraphMAE (2022) | 72.0±0.6 | 75.5±0.6 | 64.1±0.3 | 60.3±1.1 | 82.3±1.2 | 76.3±2.4 | 77.2±1.0 | 83.1±0.9 | 73.9 |
| Mole-BERT (2023) | 71.9±1.6 | 76.8±0.5 | 64.3±0.2 | 62.8±1.1 | 78.9±3.0 | 78.6±1.8 | 78.2±0.8 | 80.8±1.4 | 74.0 |
| 3D InfoMax (2022) | 69.1±1.0 | 74.5±0.7 | 64.4±0.8 | 60.6±0.7 | 79.9±3.4 | 74.4±2.4 | 76.1±1.3 | 79.7±1.5 | 72.3 |
| GraphMVP (2022) | 68.5±0.2 | 74.5±0.4 | 62.7±0.1 | 62.3±1.6 | 79.0±2.5 | 75.0±1.4 | 74.8±1.4 | 76.8±1.1 | 71.7 |
| MoleculeSDE (2023) | 71.8±0.7 | 76.8±0.3 | 65.0±0.2 | 60.8±0.3 | 87.0±0.5 | **80.9±0.3** | 78.8±0.9 | 79.5±2.1 | 75.1 |
| MoleBlend (2024) | 73.0±0.8 | 77.8±0.8 | 66.1±0.0 | 64.9±0.3 | 87.6±0.7 | 77.2±2.3 | 79.0±0.8 | 83.7±1.4 | 76.2 |
| MolCoMA (ours) | **73.5±0.5** | **78.5±0.3** | **67.4±0.4** | **65.4±0.4** | 83.8±1.3 | 79.9±0.7 | **80.6±0.6** | **85.2±0.6** | **76.8** |

The best and second best results are marked by **bold** and underlined.

## 4.3 QUANTUM PROPERTY PREDICTION TASK (3D)

We utilize the QM9 dataset (Ramakrishnan et al., 2014) to evaluate our model on molecular tasks with the 3D data format. QM9 is a dataset with 134K molecules consisting of 9 heavy atoms. It has 12 tasks related to quantum properties, including the energetic, electronic, and thermodynamic properties of molecules. Following (Thölke & Fabritiis, 2022) , we randomly select 10,000 molecules for validation and 10,831 molecules for testing. The remaining molecules are used to fine-tune our model. The results are presented in Table 2. Our model achieves state-of-the-art performance on 9 out of 12 tasks, with substantial margins in certain cases (e.g., G298, U298, U0), showcasing our model's robust capability for 3D tasks.

Table 2: Results of quantum property prediction tasks on QM9. The evaluation is Mean Absolute Error (MAE, lower is better).

| Pretraining | Alpha↓ | Gap↓ | HOMO↓ | LUMO↓ | Mu↓ | Cv↓ | G298↓ | H298↓ | R2↓ | U298↓ | U0↓ | Zpve↓ |
|---|---|---|---|---|---|---|---|---|---|---|---|---|
| Distance Prediction (2022) | 0.065 | 45.9 | 27.6 | 23.3 | 0.031 | 0.033 | 14.83 | 15.81 | 0.248 | 15.07 | 15.01 | 1.837 |
| 3D InfoGraph (2020) | 0.062 | 46.0 | 29.3 | 24.6 | 0.028 | 0.030 | 13.93 | 13.97 | 0.133 | 13.55 | 13.47 | 1.644 |
| 3D InfoMax (2022) | 0.057 | 42.1 | 25.9 | 21.6 | 0.028 | 0.030 | 13.73 | 13.62 | 0.141 | 13.81 | 13.30 | 1.670 |
| GraphMVP (2022) | 0.056 | 42.0 | 25.8 | 21.6 | 0.027 | 0.029 | 13.43 | 13.31 | 0.136 | 13.03 | 13.07 | 1.609 |
| MoleculeSDE (2023) | 0.054 | 41.8 | 25.7 | 21.4 | 0.026 | 0.028 | 13.07 | 12.05 | 0.151 | 12.54 | 12.04 | 1.587 |
| MOLEBLEND (2024) | 0.060 | **34.8** | 21.5 | 19.2 | 0.037 | 0.031 | 12.44 | 11.97 | 0.417 | 12.02 | 11.82 | 1.580 |
| SE(3)-DDM (2023) | 0.046 | 40.2 | 23.5 | 19.4 | 0.015 | 0.024 | 7.65 | 7.09 | 0.122 | 6.99 | 6.92 | 1.31 |
| 3D-EMGP (2023) | 0.057 | 37.1 | **21.3** | 18.2 | 0.020 | 0.026 | 9.30 | 8.70 | **0.092** | 8.60 | 8.60 | 1.38 |
| MolCoMA (ours) | **0.043** | 37.0 | 23.6 | **17.0** | **0.010** | **0.022** | **6.87** | **6.15** | 0.117 | **5.63** | **5.51** | **1.24** |

The best and second best results are marked by **bold** and underlined.

## 4.4 ABLATION STUDIES

In this subsection, we conduct a series of experiments to investigate the key designs of our method. First, we explore the impact of different masking strategies, including random, uniform and complementary masking strategies. Besides, we compare the performance of multi-modal and single-modal information respectively.

**Complementary masking strategy.** To explore how masking strategy impacts information interaction between different modalities, we compare complementary masking to two traditional masking strategies: random and uniform. For random masking strategy, 2D and 3D patterns are masked through individual and random sampling. For uniform masking strategy, both modalities are masked following the same pattern, such that the masked portions in one modality correspond to the masked regions in the other. In contrast, complementary sampling is quite the opposite approach.

We adopt the above three strategies in the pretraining phase and evaluate their performance after fine-tuning on MoleculeNet and QM9. As shown in Table 3, masking tokens complementarily between modalities performs the best on all tasks, which indicates that complementary masking strategy enhances molecular representation. Compared to random masking, complementary masking enables more cross-modal interactions between atoms with diverse semantic information, thus helping our model transfer knowledge between both modalities. However, with the uniform masking strategy, the extracted point cloud features and topology features are semantically aligned, so the interaction does not help the model utilize information from the other modality better.

Table 3: Comparison of cross-modal masking strategies (ROC-AUC, ↑ and MAE, ↓) on MoleculeNet and QM9.

| Masking Strategy | MoleculeNet | | | | QM9 | | | |
|---|---|---|---|---|---|---|---|---|
| | BBBP↑ | ToxCast↑ | SIDER↑ | Bace↑ | Alpha↓ | U298↓ | G298↓ | Zpve↓ |
| Random | 72.4 | 65.8 | 63.3 | 83.5 | 0.045 | 6.09 | 7.57 | 1.34 |
| Uniform | 72.5 | 66.7 | 63.5 | 82.5 | 0.047 | 6.03 | 7.15 | 1.30 |
| Complementary | **73.5** | **67.4** | **65.4** | **85.2** | **0.043** | **5.63** | **6.87** | **1.24** |

**Multi-modal v.s. Single-modal.** To demonstrate the effectiveness and cruciality of multi-modal information, we compare multi-modal and single-model in pretraining process. We only adopt 2D (or 3D) data as input to the same architecture that our method proposes in this paper. As shown in Table 4, model that leverages multiple modalities outperforms single-modal methods by a significant margin, thereby demonstrating that multiple modalities can learn more comprehensive and complementary information.

Table 4: Comparison of multi-modal and single-modal (ROC-AUC, ↑ and MAE, ↓) on MoleculeNet and QM9.

| Input | MoleculeNet | | | | QM9 | | | |
|---|---|---|---|---|---|---|---|---|
| | BBBP↑ | ToxCast↑ | SIDER↑ | Bace↑ | Alpha↓ | U298↓ | G298↓ | Zpve↓ |
| 2D | 72.3 | 65.7 | 62.1 | 82.3 | - | - | - | - |
| 3D | - | - | - | - | 0.057 | 6.24 | 7.61 | 1.53 |
| 2D+3D | **73.5** | **67.4** | **65.4** | **85.2** | **0.043** | **5.63** | **6.87** | **1.24** |

## 5 CONCLUSION

We propose MolCoMA, a self-supervised pretraining framework that fosters fine-grained 2D and 3D interaction at the atom-level. Our method achieves deep and granular interaction between modalities by masking, replacing, and recovering atom embeddings. Through cross-modal reconstruction, MolCoMA is constrained to learn the transformation information between modal data, thereby capturing the correlation and complementarity among different modal data. From the above two aspects, our method achieves state-of-the-art performance across a broad range of 2D and 3D benchmarks, demonstrating its effective and highly interactive multi-modal learning pipeline, which excels in feature extraction for both geometry and topology.

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

# A IMPLEMENTATION DETAILS OF MOLCOMA

## A.1 HYPERPARAMETERS SETTINGS

In our pretraining stage, we utilize GIN (Xu et al., 2019) as the 2D encoder backbone and TorchMD-NET (Thölke & Fabritiis, 2022) as the 3D encoder backbone. The hyperparameters associated with the network structure and pretraining process can be found in Table 5.

Table 5: Hyperparameters for MolCoMA.

| Hyperparameter | Value |
|---|---|
| Epochs | 100 |
| Batch size | 512 |
| Optimizer | AdamW |
| Adam betas | (0.9, 0.999) |
| Max learning rate | 1e-4 |
| Warm up epochs | 10 |
| Learning rate schedule | Cosine |
| masking ratio | 0.5 |
| $\alpha_1, \alpha_2, \alpha_3$ | 1 |
| 2D encoder layers number | 5 |
| 2D encoder embedding dimension | 256 |
| 3D encoder layers number | 8 |
| 3D encoder attention head number | 8 |
| 3D encoder embedding dimension | 256 |
| Unified Encoder layes number | 4 |
| Unified Encoder attention head number | 4 |
| Unified encoder embedding dimension | 128 |

Following previous methods, we adopt grid search to find the optimal hyperparameters for tasks in MoleculeNet. The search space for each task is detailed in Table 6. We manually select pretraining and finetuning hyperparameters for QM9. The hyperparameters are presented in Table 7.

Table 6: Search space for MoleculeNet tasks.

| Hyperparameter | Value |
|---|---|
| Learning rate | [1e-4, 1e-3] |
| Batch size | {8,16,32,64,128} |
| Epochs | {25,50,75,100} |
| Weight Decay | 0 |

Table 7: Hyperparameters for finetuning QM9.

| Parameter | Value |
|---|---|
| Train/Val/Test Splitting | 110000/10000/remaining data |
| Optimizer | AdamW |
| Max learning rate | 4e-4 |
| Batch size | 128 |
| Warmup Steps | 10000 |

## A.2 MOLECULE FEATURIZATION

We employ the rich atom feature strategy to featurize the molecule. The atom and bond features are shown in Table 8. Note that such atom features are only available for the 2D graph. For 3D geometry, only the atom type information is available.

Table 8: Node and edge features.

| Featurization | Hyperparameter | Value |
|---|---|---|
| Atom | Atom Type | 1~118 |
| | Atom Chirality | {unspecified, tetrahedralcw, tetrahedralccw, other} |
| | Atom Degree | 0~10 |
| | Formal Charge | -5~5 |
| | Number of Hydrogen | 0~8 |
| | Number of Unpaired Electrons | 0~4 |
| | Hybridization | {sp, sp2, sp3, sp3d, sp3d2} |
| | Is Aromatic | {false, true} |
| | Is In Ring | {false, true} |
| Bond | Bond Type | {single, double, triple, aromatic} |
| | Bond Stereotype | {none, Z variant, E variant, Cis, Trans, any} |
| | Is conjugated | {false, true} |

## B ABLATION STUDY ON MASKING RATIO

In this section, we examine the masking ratio for MolCoMA. As shown in Table 9, our model achieves best performance when the masking ratio is set 50%. Moreover, results obtained by other masking ratios also surpass most methods compared in Table 1 and Table 2, demonstrating the effectiveness of our model in fostering fine-grained interactions between modalities via a complementary masking strategy and unified encoder.

Table 9: Ablations on the masking ratio.

| Masking Ratio | MoleculeNet | | | QM9 | |
|---|---|---|---|---|---|
| | BBBP↑ | SIDER↑ | Bace↑ | Alpha↓ | G298↓ |
| 60% | 73.3 | **65.5** | 85.1 | 0.045 | 6.91 |
| 50% | **73.5** | 65.4 | **85.2** | **0.043** | **6.87** |
| 40% | 73.0 | 65.2 | 84.8 | **0.043** | 6.89 |

