# OpenReview forum: "MolCoMA: Complementary Masking Strategy for Promoting Atom-Level Multi-Modal Molecular Representation"
_ICLR.cc/2025/Conference — Submitted to ICLR 2025_

### Official Review · Reviewer_do3H · 2024-10-30

**Soundness:** 2
**Presentation:** 3
**Contribution:** 2
**Rating:** 3
**Confidence:** 4

**Summary:**

This paper proposes a novel molecular representation learning framework, termed MolCoMA, which demonstrates promising performance on the MoleculeNet and QM9 datasets. MolCoMA leverages complementary information from both topological and geometric representations while facilitating cross-modal interaction. Notably, it employs a well-designed masking technique that obscures atoms across different modalities in a complementary manner. By integrating various types of molecular information, the proposed method effectively addresses a range of downstream tasks, including biological and quantum applications.

**Strengths:**

1. The proposed method interacts the 2-D information and 3-information, which is reasonable.
2. The proposed method achieves promising results in different tasks.
3. The paper is well-written and easy to follow.

**Weaknesses:**

1. Lack of novelty. The interaction between 2d-topology and 3d-geometry has already the explored in previous studies, such as GraphMVP [a], Transformer-M [b].

2. MolCoMA is pre-trained on PCQM4Mv2, while Mole-Bert is pre-trained on ZINC and GraphMVP is pre-trained on GEOM. Therefore, the comparison is not fair. To verify the effectiveness of the proposed method, the pre-training datasets of different methods should be the same.

3. More benchmarks on 3D tasks should be conducted, such as MD17 and LBA. Moreover, the proposed method should be compared with recent works SLIDE [c] and Frad [d].

4. Why is complementary masking better than other strategies? The masked information of one modality can be revealed by another modality. The authors should conduct more experiments or theoretical analyses between different masking strategies and verify why complementary masking is effective.

5. To verify the effectiveness of the MolCoMA, it would be better to conduct experiments on more architectures such as EGNN [e]

[a] PRE-TRAINING MOLECULAR GRAPH REPRESENTATION WITH 3D GEOMETRY, ICLR 2022.

[b] ONE TRANSFORMER CAN UNDERSTAND BOTH 2D & 3D MOLECULAR DATA, ICLR 2023.

[c] Sliced Denoising: A Physics-Informed Molecular Pre-Training Method, ICLR 2024.

[d] Fractional Denoising for 3D Molecular Pre-training, ICML 2024.

[e] E(n) Equivariant Graph Neural Networks, ICML 2021.

**Questions:**

Overall, the contribution of this paper seems incremental and the experimental evaluations are not fully convincing in several aspects. Please refer to weaknesses for details.

---

### Official Review · Reviewer_djZX · 2024-11-03

**Soundness:** 3
**Presentation:** 2
**Contribution:** 2
**Rating:** 5
**Confidence:** 4

**Summary:**

This paper proposes a multi-modal molecular representation pre-training framework, MolCoMA, to facilitate fine-grained interactions of intrinsic features across modalities. The framework employs two modality-specific encoders to capture unique characteristics of each modality and a unified encoder with a complementary masking mechanism to integrate these features. By optimizing in-modal and cross-modal reconstruction losses, this framework learns robust 2D and 3D molecular representations.

**Strengths:**

1. The motivation is understandable and reasonable, as previous work on multi-modal molecular representation learning has overlooked the cross-modal complementarity at the atom level.
2. The proposed method is technically sound, as similar approaches have already been proposed and validated in image-text multi-modal representation pre-training.
3. The experimental results on MoleculeNet and QM9 look promising, and the proposed method has been compared with the latest baseline models.

**Weaknesses:**

1. The datasets used for evaluation in the experiments are relatively limited. For 2D, the evaluation is only conducted on classification datasets from MoleculeNet, without testing on regression datasets. For 3D, the evaluation is only conducted on QM9, without testing on datasets like MD17 or GEOM-Drugs.
2. The Unified Encoder is a vanilla self-attention module that does not take structural inductive bias into account.
3. Some details of the method are not clearly described, such as the implementation of the cross-modal reconstruction function in Equation 8.

**Questions:**

1. Can the attention map of the Unified Encoder be visualized to examine the interactions between modalities?
2. Can the outputs of two unique encoders, which are not in the same representation space, be directly merged and encoded together without alignment?
3. What impact does the structure of the GNN Tokenizer used as a 2D Decoder have on the results? For example, GCN, GAT, GIN, Graph Transformer.
4. What impact do the three sub-objectives in Equation 9 have on performance individually? It would be helpful to provide an ablation analysis of the three sub-objectives.

---

### Official Review · Reviewer_6Joc · 2024-11-08

**Soundness:** 3
**Presentation:** 2
**Contribution:** 2
**Rating:** 5
**Confidence:** 4

**Summary:**

This paper introduces MolCoMA, a molecular representation learning framework that leverages two modalities of molecules: 2D topology and 3D geometry. Specifically, MolCoMA pretrains 2D and 3D molecule encoders (with another unified encoder) through masked auto-encoding with complementary masking. This masking strategy encourages the model to learn modality-specific characteristics more effectively, filling in the gaps between the complementarily-masked modalities. As a result, the pre-trained 2D and 3D encoders via MolCoMA achieve state-of-the-art performance on 2D downstream tasks (MoleculeNet) and 3D downstream tasks (QM9).

**Strengths:**

- MolCoMA achieves strong performance in two standard molecule benchamrks, MoleculeNet and QM9.
- The idea of unified encoding with complementary masking is simple yet effective. Although I think this is not new in the self-supervised learning community, e.g., [1-2], but the idea has been under-explored in the molecule domain yet, so it is somewhat novel.

[1] MixMAE: Mixed and Masked Autoencoder for Efficient Pretraining of Hierarchical Vision Transformers \
[2] Mixed Autoencoder for Self-supervised Visual Representation Learning

**Weaknesses:**

1. The method is not comprehensively described.
   - What are the data transformations, $T\_\text{2D}$ and $T\_\text{3D}$?
   - Is the unified encoder a vanilla bidirectional transformer? Does it simply take the mixed representations as an input sequence?
   - Is the output sequence of the unified encoder directly passed into the 2D and 3D decoders?
   - What is the Simple GNN Tokenizer? Is it randomly initialized or a pretrained model? Due to the stop-gradient operation, the tokenizer is not optimized, so it requires a detailed description.
   - When is the complementary masking applied? In Figure 1, it appears that masking is applied before modality-specific encoding. However, in Section 3, it seems that masking is applied right before unified encoding.
   - In cross-modal reconstruction, $h\_\text{2D}$ and $h\_\text{3D}$ appear to be modality-specific representations that do not rely on the unified encoder. Is this correct? (I assume $h\_\text{2D}$ and $h\_\text{3D}$ are only defined in L252-L263).
   - What is meant by avoiding the use of visible geometric tokens in L333? It is difficult to understand how cross-modal reconstruction works and why visible tokens should be avoided.
   - Overall, there are several confusing notations, and many parts are not clearly explained. I strongly recommend the authors use concrete mathematical notations to reduce confusion.
2. Some concerns on experimental results.
   - Some baselines use different pretraining datasets compared to this paper. For example, MoleBERT uses 2M molecules from the ZINC15 database, and 3D-EMGP and GraphMVP are pretrained on the GEOM dataset containing 50K–100K molecules. However, MoleCoMA uses PCQM4Mv2 with 3.4M molecules. This difference in pretraining datasets could significantly impact performance, so I wonder if MoleCoMA is still effective with a different (smaller) dataset.
   - The paper lacks analysis of the effectiveness of the unified encoder. Since the authors claim the unified encoder as a contribution of this paper, providing such analysis is crucial.
   - One way to interpret the quality of molecular representations is molecule retrieval based on the learned representations. If the retrieval finds chemically similar molecules well, one could infer that the learned representations are chemically informative and useful for downstream tasks.

**Questions:**

- What is the difference between $H(l)$ and $H^{(l)}$ in Eq (4)?
- How to choose the hyperparameters provided in Table 5? Since molecular representation learning frameworks are often sensitive to the hyperparameters, the hyperparameter search strategy is also important.

---

### Official Review · Reviewer_WAKa · 2024-11-09

**Soundness:** 1
**Presentation:** 2
**Contribution:** 1
**Rating:** 3
**Confidence:** 4

**Summary:**

The paper proposes MolCoMA, a self-supervised molecular pretraining framework that integrates information from both 2D topology and 3D geometry in a fine-grained manner. The architecture is consisted of two modality-specific encoders, leading to a unified cross-modal encoder that leverages a complimentary masking strategy to mitigate representational overlap between the two modalities. 2D and 3D modalitity-specific decoders then perform feature reconstruction and conformer denoising, respectively, in addition to a cross-modal node-wise feature reconstruction task that aims to recover 2D features from the 3D features. Experiments on MoleculeNet and QM9 datasets show that MolCoMA outperforms previous work on diverse molecular property prediction tasks.

**Strengths:**

- [S1] **Simplicity of approach.** Each component of MolCoMA is simple and is presented clearly such that the paper is easy-to-follow.
- [S2] **Interesting Topic.** Pretraining GNNs that are generalizable to various molecular property prediction tasks is a well-studied problem, and MolCoMA could be a good addition to try in the molecular pretraining literature for practitioners.

**Weaknesses:**

- **Unclear motivation and intuition.** Despite complementary masking being the main contribution of this paper, the intuition behind the approach is hard to understand.
  - [W1] In general, a more difficult pretraining task is expected to bake in more useful knowledge into neural networks as seen with using hard negatives in contrastive learning literature [A]. Based on this intuition, it seems counterintuitive that complementary masking outperforms random/uniform masking as the ground-truth targets can easily be found based on nodes from the other modality during pretraining. In other words, how can we expect the 2D-specific encoder to perform well on downstream tasks with no 3D conformers, if it was pretrained towards relying on 3D-information, which is not present during finetuning?
  - [W2] Throughout the paper, the authors mention 2D and 3D information "each possess unique representational strengths" (Line 210), yet claim that uniform/random masking used in previous work "result in feature redundancy due to the overlap" (Line 207), which makes the main motivation for complementary masking self-contradictory. If 2D and 3D information indeed contain unique information, shouldn't it be the case that random masking should be enough since they express different knowledge? But if there is large representational overlap, that means complementary masking makes the task too easy for the model, which brings us to the point of W1 above. Either way, the intuition behind the proposed method needs further discussion and clarification.

- **Insufficient justification on method design choices.** The proposed architecture and objective are not justified properly and requires further discussion.
  - [W3] The unified encoder composed of vanilla Transformer layers fails to preserve symmetry within the data distribution, which is crucial in ensuring generalizability and robustness. Specifically, SE(3) transformations $g(\cdot)$ on the 3D geometry $\mathbf{R}$ are not respected for 3D noise prediction (i.e., $f_{\mathbf{\theta}}( \mathbf{X}, \mathbf{E}, g(\mathbf{R}) ) \neq g(f_{\mathbf{\theta}}( \mathbf{X}, \mathbf{E}, \mathbf{R})$) as vanilla attention is not equivariant to SE(3) roto-translations. This equivariance property could be enforced (1) approximately via data augmentation [B] or (2) exactly by replacing the attention mechanism [C], but MolCoMA discusses neither of these, making the architecture design less reliable.
  - [W4] The cross-modal reconstruction objective seems ill-defined, in the sense that the model is trained to map 3D features that are responsible for predicting the ground-truth noise (thereby depends on the noise added to the 3D conformer), to 2D features that are stable regardless of the noise. In effect, this could result in a suboptimal trade-offs for the 3D denoising task, yet this discussion is only done vaguely in Lines 324-332.
  - [W5] Lastly, the final objective (Equation 9) involves a weighted sum of three distinct loss functions, without any guidance on how the weights should be set. It would be interesting to test how the performance varies with different weights (other than $\alpha_1 = \alpha_2 = \alpha_3 = 1$ case), possibly leading to insights on how each modality contributes to molecular property prediction.

[A] Robinson et al., Contrastive Learning with Hard Negative Samples. ICLR 2021.\
[B] Quiroga et al., Revisiting Data Augmentation for Rotational Invariance in Convolutional Neural Networks. AISC 2019.\
[C] Fuchs et al., SE(3)-Transformers: 3D Roto-Translation Equivariant Attention Networks. NeurIPS 2020.

**Questions:**

- [Q1] When fusing the graph and geometry features prior to feeding them to the unified encoder, is there any mechanism used to distinguish tokens from the 2D graph and those from the 3D geometry?
- [Q2] The overall pipeline highly resembles MultiMAE [B] in computer vision, where a non-uniform masking strategy was used by sampling probabilities from a Dirichlet distirbution, then masking each modality with its corresponding probability. Have the authors considered non-uniform masking probabilities as such?
- Small typo in Line 271: did you mean "concatenate" instead of "contact"?

[D] Bachmann et al., MultiMAE: Multi-modal Multi-task Masked Autoencoders. ECCV 2022.

---

### Meta-Review · Area_Chair_b96z · 2024-12-19

**Metareview:**

The paper introduces MolCoMA, a multi-modal molecular pretraining framework that aims to integrate both 2D topological and 3D geometric information at the atom-level. The core idea is to use a complementary masking strategy to ensure that for each atom (node), certain features are masked in one modality and preserved in the other, encouraging fine-grained, cross-modal feature interaction.

The paper received negative scores from all reveiwers. The primary concerns from the reviewers were about (1) unclear motivation, since it is unclear how the proposed complementary masking strategy improves the pre-training over random masking, (2) insufficient justifications on model designs, since the unified encoder as a vanilla Transformer does not respect symmetry or equivariance constraints, (3) issues in evaluation, since the proposed model is pretrained on a larger and different dataset (PCQM4Mv2) compared to certain baselines, and (4) limited benchmarks, since the results are mostly focused on a classification on MoleculeNet dataset.

Since the authors did not provide the rebuttal, AC agrees with the reviewers’ decision and recommends rejection.

**Additional Comments On Reviewer Discussion:**

The reviewers raised four critical concerns discussed in the meta-review. Since the authors did not provide a rebuttal, the reviewers maintained the recommendations.

---

### Decision · Program_Chairs · 2025-01-22

Reject